# Synthesizing Complex-Valued Multicoil MRI Data from Magnitude-Only Images

**DOI:** 10.3390/bioengineering10030358

**Published:** 2023-03-14

**Authors:** Nikhil Deveshwar, Abhejit Rajagopal, Sule Sahin, Efrat Shimron, Peder E. Z. Larson

**Affiliations:** 1UC Berkeley-UCSF Graduate Program in Bioengineering, Berkeley, CA 94701, USA; 2Department of Radiology and Biomedical Imaging, University of California, San Francisco, CA 94016, USA; 3Department of Electrical Engineering and Computer Sciences, University of California, Berkeley, CA 94701, USA

**Keywords:** synthetic phase, synthetic multi-coil data, deep generative models, GANs, generative adversarial network, synthetic data, MRI reconstruction, deep learning, unrolled networks

## Abstract

Despite the proliferation of deep learning techniques for accelerated MRI acquisition and enhanced image reconstruction, the construction of large and diverse MRI datasets continues to pose a barrier to effective clinical translation of these technologies. One major challenge is in collecting the MRI raw data (required for image reconstruction) from clinical scanning, as only magnitude images are typically saved and used for clinical assessment and diagnosis. The image phase and multi-channel RF coil information are not retained when magnitude-only images are saved in clinical imaging archives. Additionally, preprocessing used for data in clinical imaging can lead to biased results. While several groups have begun concerted efforts to collect large amounts of MRI raw data, current databases are limited in the diversity of anatomy, pathology, annotations, and acquisition types they contain. To address this, we present a method for synthesizing realistic MR data from magnitude-only data, allowing for the use of diverse data from clinical imaging archives in advanced MRI reconstruction development. Our method uses a conditional GAN-based framework to generate synthetic phase images from input magnitude images. We then applied ESPIRiT to derive RF coil sensitivity maps from fully sampled real data to generate multi-coil data. The synthetic data generation method was evaluated by comparing image reconstruction results from training Variational Networks either with real data or synthetic data. We demonstrate that the Variational Network trained on synthetic MRI data from our method, consisting of GAN-derived synthetic phase and multi-coil information, outperformed Variational Networks trained on data with synthetic phase generated using current state-of-the-art methods. Additionally, we demonstrate that the Variational Networks trained with synthetic *k*-space data from our method perform comparably to image reconstruction networks trained on undersampled real *k*-space data.

## 1. Introduction

Deep learning-based MRI reconstruction methods show promise in faithfully reconstructing MR images from undersampled *k*-space measurements, but such methods are usually hampered by a lack of paired and diverse training data, posing a barrier to effective clinical translation of these technologies. Current deep learning MRI reconstruction techniques use datasets [1,2,3,4,5,6,7,8,9,10,11,12,13] containing paired images and raw *k*-space MRI data and have enabled major advances in MRI reconstruction methods. However, they are limited in several ways. Magnitude images contained in these datasets are sometimes preprocessed which can lead to biased results for MRI reconstruction [14] and are hard to standardize. Furthermore, these publicly available datasets are typically limited in anatomy, acquisition parameters and pathology information. Recent studies have shown that such limitations can sometimes result in hallucinations of structures or artifacts during deep learning-based MRI reconstruction [15,16], limiting the generalization potential of these methods and their clinical use.

There could be significant advantages to leveraging the diversity of existing clinical MRI databases as they contain a range of patient populations, anatomy, pathology, image contrasts, acquisition parameters, and data from different vendors. This would be particularly useful for multi-task networks, e.g., [17], that perform both image reconstruction and a downstream task such as segmentation or classification. Training on more diverse and representative datasets can also greatly contribute to improving deep learning reconstruction models, especially for rare anatomies and pathologies; this could potentially allow for greater clinical adoption.

However, we cannot simply use clinical datasets for MRI reconstruction algorithm development because they typically only contain magnitude images while image phase information is discarded. Furthermore, MRI data are acquired from multi-channel RF coils, but clinical images show a coil-combined image and, thus, the multi-channel information is lost. MRI phase data are important because they contain information related to contrast from chemical shift, magnetic susceptibility differences, inhomogeneities in the main magnetic field, RF coils used, fat/water separation, tissue interfaces, blood flow, and temperature change [18,19,20,21,22,23]. Additionally, recent studies have shown that using complex-valued neural networks which operate on data that include phase information produce higher quality reconstructed images [24,25].

Thus, the ability to recover or generate image phase from already completed scans could increase the utility and applicability of deep learning MRI reconstruction methods. While a variety of techniques aim to synthesize different MRI contrasts [26,27,28,29,30,31] or parameter maps [32,33], relatively few techniques exist to synthesize MR image phase and complex-valued multi-coil data. Recent studies have included methods to generate synthetic image phase by emulating very specific physical models [34], generating sinusoidal phase [35], or have focused on fine tuning training datasets consisting mostly of natural images [36]. To the best of our knowledge, no methods have attempted to broadly synthesize realistic MRI phase maps.

To address this, we present a method for synthesizing realistic MRI data, including image phase and multi-channel information, from magnitude-only images that, for example, are found in clinical imaging archives. Our method leverages recent advances in deep generative modeling [37,38] to generate synthetic MRI phase images from input MRI magnitude images. Corresponding coil sensitivity maps are derived and then used to generate synthetic multi-channel data. The resulting synthetic multi-coil MRI data, including synthesized image phase, were then evaluated for their ability to be used in image reconstruction tasks by training a Variational Network [39] and comparing to a network trained on real multi-coil MRI data. Our results show that the proposed method (i) generates realistic looking MR phase maps, (ii) outperforms current methods used to generate synthetic phase data for training reconstruction models and (iii) image reconstruction networks trained on synthetic multi-coil data perform comparably to the same networks trained on real data. Our findings suggest that this framework has the potential to address the limitations that exist in current MRI datasets used for reconstruction tasks where access to raw *k*-space data is required.

## 2. Methods

We first start by defining *k*-space, magnitude, and phase in mathematical terms. We then describe generating synthetic phase images from input magnitude-only images using a conditional generative adversarial network (GAN) framework. Finally, we describe the evaluation of the synthetic data quality.

### 2.1. Preliminaries

The signal acquired from a 2D slice (assuming we can neglect $T_{2}$ decay) in the spatial frequency domain, or *k*-space, can be expressed as:(1)$$M\left(k_{x},k_{y}\right)=\int_{X}\int_{Y}m\left(x,y\right)e^{-j2\pi (k_{x}x+k_{y}y)}dxdy,$$
where m(x,y) is the signal generated at the position (x,y). This is a complex quantity which is equivalent to
(2)$$m\left(x,y\right)=m_{x}\left(x,y\right)+jm_{y}\left(x,y\right),$$
where $m_{x}\left(x,y\right)$ is the real component of the signal and $m_{y}\left(x,y\right)$ is the imaginary component of the signal. The goal of MRI reconstruction is to recover m(x,y) from $M(k_{x},k_{y})$.

The MR signal, m(x,y), is generated by the rotation of the transverse components of the net magnetization. The signal is complex-valued because it is a measurement of both the *x* and *y* components of the net magnetization. The majority of MRI scans are interpreted based on the magnitude of the signal, |m(x,y)|, which corresponds to the amplitude of the transverse magnetization. There is also information encoded in the phase (also known as angle) of the signal, ∠m(x,y), which corresponds to the rotation angle of the transverse magnetization. This includes chemical shift, magnetic susceptibility differences, inhomogeneities in the main magnetic field, RF coil profiles, fat/water separation, tissue interfaces, and blood flow.

### 2.2. Generative Modeling

#### 2.2.1. Neural Network Architecture

The generator is a 16-layer U-Net [40] with skip connections and the discriminator is a 70 × 70 PatchGAN [38]. In this setup, the discriminator, in a convolutional manner, decides if a patch is real or fake. We used a PatchGAN discriminator to restrict the network’s attention to the structure of local image patches. This encourages the discriminator to penalize the structure at the scale of patches rather than the whole image (as in a typical binary classifier) in order to effectively capture high-frequencies in the synthetic image. In a sense, PatchGAN acts as a classifier itself. The main difference is that the output of the PatchGAN is an *N* × *N* array in which each element signifies whether the corresponding patch in the image is real or fake. We chose a patch size of 70 × 70 based on the results of previous studies [38], which empirically found that this patch size gives the best tradeoff between image sharpness and alleviating artifacts in the generated image. Each generative model was trained with a batch size of 1. We used minibatch stochastic gradient descent (SGD) with the Adam optimizer [41] using a learning rate of 2 × 10${}^{-4}$ and momentum parameters $\beta_{1}=0.5$ and $\beta_{2}=0.999$. Additional implementation details can be found in [38].

#### 2.2.2. Synthetic Phase Generation

The conditional GAN uses a hybrid objective consisting of two loss functions: a conditional adversarial loss function and a regularized *ℓ*1 distance loss function. In essence, we trainined the model to generate high-frequency structures in the synthetic image, and we used the *ℓ*1 loss to control how many low-frequency structures were present in the image.
(3)$$G^{*}=arg\min_{G}\max_{D}\underset{L_{cGAN}}{\underbrace{E_{x,y}\left[logD\left(x,y\right)\right]+E_{x,z}[log\left(1-D\left(x,G\left(x,z\right)\right)\right]}}+\lambda \underset{L_{\ell 1}}{\underbrace{E_{x,y,z}{[∥y-G\left(x,z\right)∥}_{1}]}},$$
where *x* is the input magnitude image, *y* is the generated synthetic phase image that corresponds to *x*, and *z* is the latent vector. In this objective, *G* tries to minimize the objective while *D* tries to maximize it. This setup is suitable for our aim because the discriminator is conditioned on the input image *x*, and we have access to the raw ground-truth data, and thus also to the ground-truth phase data. The network was optimized by alternating between gradient descent steps conducted for optimizing the discriminator and the generator, similar to the approach as described in the original GAN paper [37].

Specifically, we trained a U-Net to predict the phase component from input magnitude-only images. During training, this synthetic phase component was compared to the ground truth phase using a hybrid objective (Equation 3). This mixed loss function balances realistic looking phase images via the adversarial loss and encourages less blurring via the *ℓ*1-norm. Each GAN model was trained for 50 epochs with a qualitative analysis of realistic synthetic phase maps being the main stopping criteria. During inference, the trained U-Net was used to generate synthetic phase from previously unseen magnitude images, resulting in the creation of realistic synthetic MRI phase data.

#### 2.2.3. Multi-Coil Data Generation

To generate synthetic multi-coil *k*-space data, we first analytically converted the input magnitude and generated synthetic phase images to real and imaginary components. Sensitivity maps were then generated using the ESPIRiT [42] algorithm on corresponding ground-truth raw data from the training dataset. The resulting sensitivity maps were multiplied with the real and imaginary components to generate multi-coil synthetic *k*-space data. This resulting complex-valued data can be used in place of ground truth *k*-space data to train deep learning based MRI reconstruction networks.

### 2.3. Dataset

Multi-coil *k*-space data obtained from the fastMRI [1] dataset were used for training the conditional GAN. The dataset consists of raw complex-valued *k*-space with both magnitude and phase information of brain scans at 1.5 T and 3 T. The images were acquired with a fast spin echo (FSE) pulse sequence with an echo train length (ETL) of 4. For training, we divided the dataset into two datasets with 16-coil and 20-coil acquisitions. Each dataset consisted of $T_{1}$-weighted, $T_{2}$-weighted, and FLAIR contrast images.

### 2.4. Experiments

#### 2.4.1. Generative

We trained two generative models: A 16-coil model and a 20-coil model, trained on 22,691 and 18,519 magnitude-only brain images, respectively. During training, the U-net generator generated a synthetic phase image and the discriminator compared the generated image to the corresponding ground truth phase image (obtained from fastMRI) in a convolutional patch-wise manner. At inference time, magnitude-only images from the fastMRI test set were run through a forward pass of the trained generative models. In our experiments, this enabled the generation of 6541 synthetic phase images for the 16-coil model and 5845 synthetic phase images for the 20-coil model.

#### 2.4.2. Evaluation: Physics-Based Image Reconstruction

To evaluate the utility of complex-valued multi-coil *k*-space data synthesized from the generative model, we compared the quality of reconstructed MR images from reconstruction networks trained on ground-truth and synthetic data. Multiple equispaced undersampling masks of acceleration factors R={4,6,8,10} with a center fraction of 0.04 were applied to *k*-space data to be used for training. Two Variational Networks [39] were then trained for 10 epochs with the 16-coil and 20-coil datasets each. Each Variational Network was trained separately on synthetic and ground truth multi-coil *k*-space with a 80/10/10 training/validation/test split for a total of four trained reconstruction networks per acceleration factor. Each trained reconstruction model (ground-truth and synthetically trained) was then run on the same ground-truth test set. The quality of reconstructed magnitude images was evaluated using standard quantitative image reconstruction metrics: PSNR, NMSE, SSIM [43]. We decided to use the Variational Network for evaluation because of its reliance on undersampled multi-coil *k*-space and coil sensitivity maps as inputs into the unrolled network. Please see the Figure 1.

All models (generative and reconstruction) were implemented in PyTorch and were trained on NVIDIA (Santa Clara, CA, USA) Titan RTX and Quadro RTX 8000 GPUs.

## 3. Results

Figure 2 shows sample comparisons between synthetic and ground truth phase images. The synthetic phase images show several expected features, including low spatial-frequency components, a noisy background, and tissue phase contrast, for example, between the ventricles and adjacent brain tissue. We do not expect the synthetic phase to exactly match the ground truth phase because the MRI phase is not deterministic and can vary based on $B_{0}$ homogeneity and RF coil induced phase shifts. In some cases, blocking artifacts have appeared, possibly due to the PatchGAN discriminator.

Figure 3 and Figure 4 show representative images reconstructed with Variational Network models trained with undersampled ground truth and synthetic *k*-space data, correspondingly. For R={4,8} acceleration factors, the reconstructed images trained on synthetic data contain slightly more error structures compared to the images trained on ground-truth data. However, visually, there are no obvious artifacts in the reconstructed images in either method. For the R=8 acceleration factor, we can see more errors in high resolution features, possibly due to the lack of high frequency details in the synthetic phase images used to train the reconstruction network.

Figure 5 and the Appendix A compare the effect of different types of phase on Variational Network reconstruction performance at various acceleration factors. The reconstruction networks were trained on ground truth data, synthetic data (from our proposed method), sinusoidal phase data, data with random phase and data with zero phase. From the plots, Variational Networks trained on undersampled synthetic data perform comparably to the same networks trained on ground truth undersampled *k*-space at R={4,6} as measured by PSNR, NMSE and SSIM. At R={8,10} acceleration factors, the performance of networks trained on synthetic data dips, especially the SSIM curve, but remains relatively comparable to that of the networks trained on ground truth data. Additionally, the networks trained on synthetic data outperform networks trained on sinusoidal phase data in all quantitative metrics for the 20-coil dataset. For the 16-coil dataset, similar results were observed for the PSNR and NMSE measurements, while the performance in the SSIM metric was comparable to the sinusoidal phase trained network.

## 4. Discussion

There is a massive amount of magnitude-only images as this is what is typically stored in clinical imaging databases (e.g., PACS), which do not usually contain phase and multi-coil information or raw *k*-space data. This work proposes a framework to generate synthetic multi-coil MRI data from magnitude-only MR images, and evaluates its utility by training a deep learning-based image reconstruction network using the synthesized datasets. The demonstrated framework aims to allow for the use of these large imaging databases for developing data-driven methods that require MRI raw data. We chose to evaluate using a Variational Network image reconstruction model as a proof of concept to demonstrate the effectiveness of the method. We believe a more significant opportunity for such a synthetic data pipeline is to train multi-task networks, e.g., networks that perform both image reconstruction and a downstream task such as image segmentation or classification [17]. In these methods, the synthetic data pipeline can take advantage of existing clinical images and annotations for the downstream tasks, enabling the creation of customized datasets for multi-task machine learning techniques.

Other approaches that generate synthetic MRI training data typically build on natural image datasets. For example, in [34], the authors simulated signal voids in MR images by randomly applying masks to natural images to generate synthetic data. Additionally, in [35,36], the authors used a natural image dataset and a magnitude-only MRI dataset, respectively, and modulated the training images with a sinusoidal phase at a random frequency. They demonstrated that training with this synthetic data showed substantially higher levels of aliasing artifacts compared to using real MRI data. The proposed generative modeling approach shows more realistic image phase maps that include both the low-frequency features, which these prior methods aimed to incorporate, as well as contrast based on the underlying tissues and anatomy (Figure 2). Our quantitative results (Figure 5) suggest that encoding this tissue phase information (not just low-frequency or sinusoidal phase information) into training data for deep learning models adds more useful information for the network to learn higher quality image reconstructions.

The authors of [36] observe that deviations in SNR, acquisition type, and aliasing patterns between training and testing times can result in widely varying image reconstruction quality. With this in mind, future experiments can extend our work to exploit the synthetic data pipeline and large clinical imaging databases to generate custom heterogeneous datasets to train more robust and generalizable image reconstruction models.

In addition to generating synthetic phase maps, a major aim of this work was to generate multi-coil data to increase the clinical relevancy of the technique. We take advantage of the well-established coil sensitivity map algorithm ESPIRiT [42] to estimate coil sensitivities instead of trying to learn them directly. This approach requires running ESPIRiT on prior ground truth data from fastMRI and, thus, a paired dataset with magnitude and ground truth phase information is still required for this part of the method for image generation.

In previous experiments, we tried to generate multi-coil data by adding a two-channel real and imaginary component to the output of the conditional GAN. This would result in generated synthetic real and imaginary images for *N* coils from a single magnitude-only image input. While this approach produced reasonable phase maps and comparable reconstructions for generative models trained on data acquired with a small number of coils (e.g., N=4), the phase maps resulting from generative models trained on N={16,20} number of coils suffered from large amounts of structure hallucination and blocking artifacts. We hypothesize that, during training, gradients across multiple individual coil images are ill-behaved and, thus, GAN models trying to generate a large number of coil images have difficulty converging.

The advantage of our proposed technique is that it is coil-agnostic; it can be applied to MR images acquired with any number of coils with the generative model learning a one-to-one mapping from magnitude to phase. This results in more stable training and gradient flow, especially for GANs. It is important to note that we do not expect the synthetic phase maps to be necessarily consistent with the ground truth phase maps. This is because MR phase is not deterministic, and can vary due to tissue composition, scan parameters such as TE, magnetic field homogeneity, and the RF coil configuration and loading. This inconsistency would be problematic for performing any quantification on the synthetic maps themselves. However, consistency with ground truth phase for individual datasets is not required when the synthetic data are used for training, but rather the synthetic phase should be consistent with population-level phase patterns. Nevertheless, enforcing a physics-based consistency between the input magnitude image and output phase image by adding a regularized term in *k*-space to the training objective could be a useful follow-up experiment to this work. Such a change could result in even more representative phase maps. However, GAN stability during training with this new objective remains an open question and would have to be answered empirically. In lieu of this, a score-based generative model could be used for this technique due to their improved training stability compared to GANs [44].

A current limitation of this study is that only fast spin-echo (FSE) images from the fastMRI database were used to train the generative model. The exclusion of gradient-recalled echo (GRE) acquisition data in the training dataset makes the trained generative models and downstream reconstruction models susceptible to distribution shift errors. To address this limitation, future work could include fine-tuning the generative models trained on FSE data with GRE data. Additionally, quantifying the uncertainty in distributions not seen at inference time as proposed in [45] could provide insight into how the generative model is synthesizing phase images on a pixel-wise basis.

Finally, the evaluation of generative models, especially for synthetic medical imaging data, is still an open research direction [46]. While this study used an unrolled image reconstruction network to evaluate the utility of the synthesized complex-valued multi-coil data, other methods, e.g., the Inception Score [47] or FID score [48], could be used to characterize the distribution of that data. Incorporating a customized implementation of these distance metrics based on medical imaging datasets [49] could be more fruitful in characterizing synthetic MR phase images. This information could also possibly be used to direct generative model training to synthesize datasets customized for specific downstream tasks.

## 5. Conclusions

This work presents a new method for synthesizing realistic, multi-coil MRI data from magnitude-only images that uses a GAN to generate image phase and ESPIRiT-generated coil sensitivity maps. The synthetic data were evaluated by comparing the reconstruction performance of Variational Networks trained on real *k*-space and synthetic *k*-space data. Our results suggest that the proposed method for generating synthetic data (i) outperforms the current state-of-the-art methods for creating synthetic image phase and (ii) is adequate for training deep learning MRI reconstruction models at typical acceleration factors (up to 10×), shown by the Variational Networks results. Taken together, our results suggest that image-to-image translation generative adversarial networks are able to generate MRI phase images that are both realistic-looking and can also provide a good performance when used for training an image reconstruction network. This allows for the possibility of using large, diverse clinical imaging databases that contain magnitude-only images when developing deep learning MRI reconstruction methods.

## Figures and Tables

**Figure 1 bioengineering-10-00358-f001:**
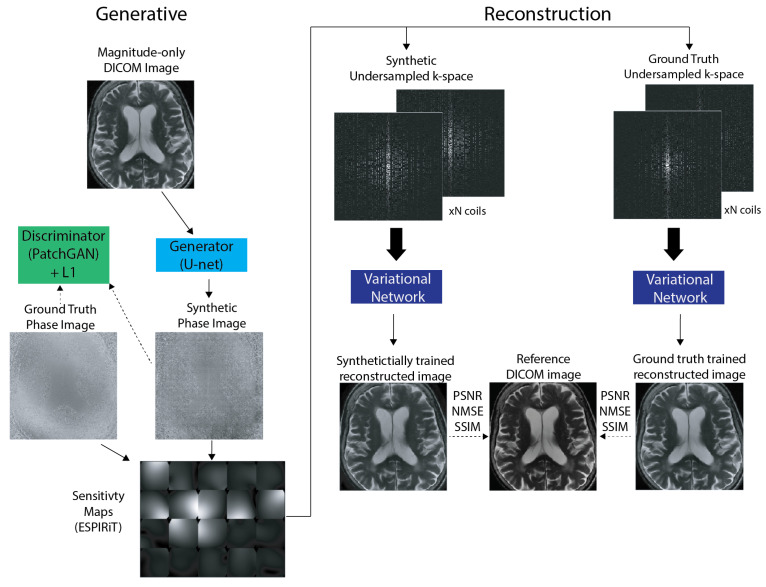
The proposed synthetic raw data generation and image reconstruction pipeline. The generative model takes magnitude images as an input seed and produces plausible synthetic phase images as output, which are trained to match ground truth phase images from the dataset. Synthetic complex-valued data is obtained by combining the input (ground truth) magnitude image and synthetic phase image to yield real and imaginary components. Estimated sensitivity maps calculated with ESPIRiT from the training dataset are then applied to synthetic complex-valued multi-coil data to compute multi-coil *k*-space encoded with synthetic phase information. The synthetic raw data were evaluated by training a Variational Network using undersampled *k*-space data.

**Figure 2 bioengineering-10-00358-f002:**
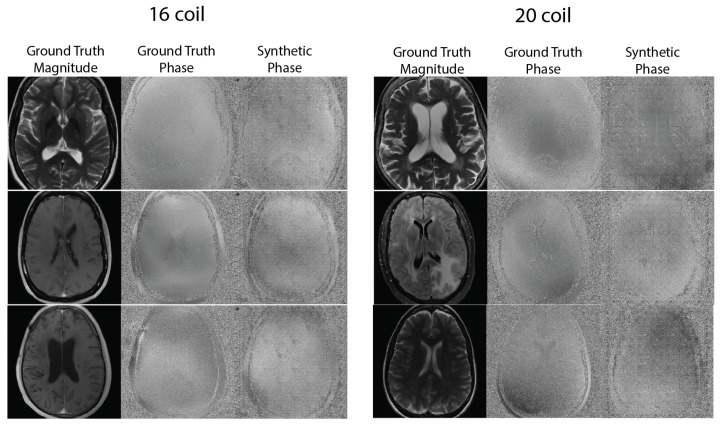
Representative ground truth magnitude, ground truth phase, and synthetic phase images generated from the conditional GAN. Synthetic phase images show expected features, including appropriate noise patterns, low spatial-frequency components and tissue contrast between the ventricles and nearby brain tissue, but exhibit some blocking artifacts possibly from the patchGAN discriminator.

**Figure 3 bioengineering-10-00358-f003:**
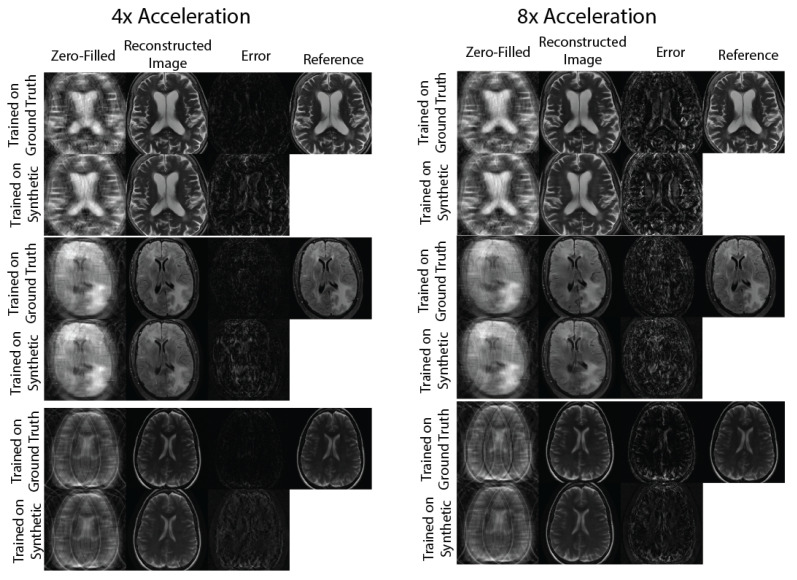
Sample image comparisons at 4× and 8× acceleration factors for the 20-coil dataset. The columns compare the zero-filled image, reconstructed image, and error maps generated with 2 Variational Networks trained on ground truth and synthetic *k*-space.

**Figure 4 bioengineering-10-00358-f004:**
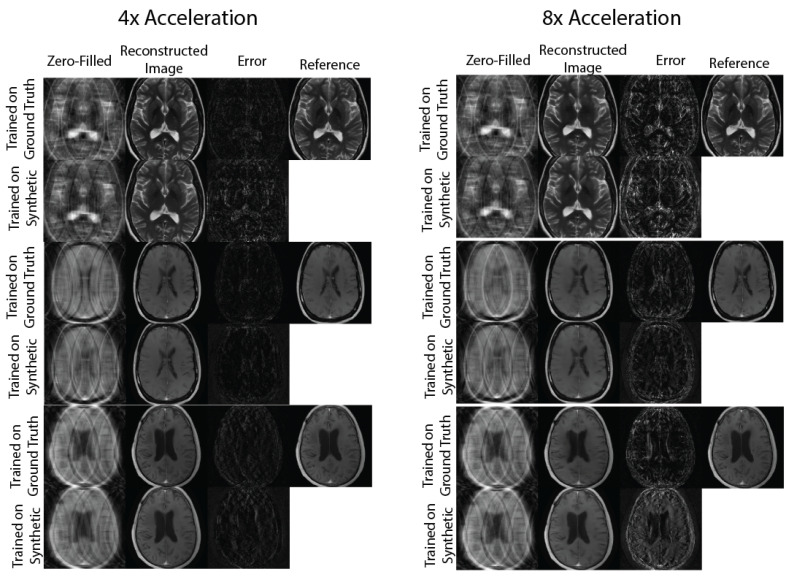
Sample image comparisons at 4× and 8× acceleration factors for the 16-coil dataset. The columns compare the zero-filled image, reconstructed image, and error maps generated with 2 Variational Networks trained on ground truth and synthetic *k*-space.

**Figure 5 bioengineering-10-00358-f005:**
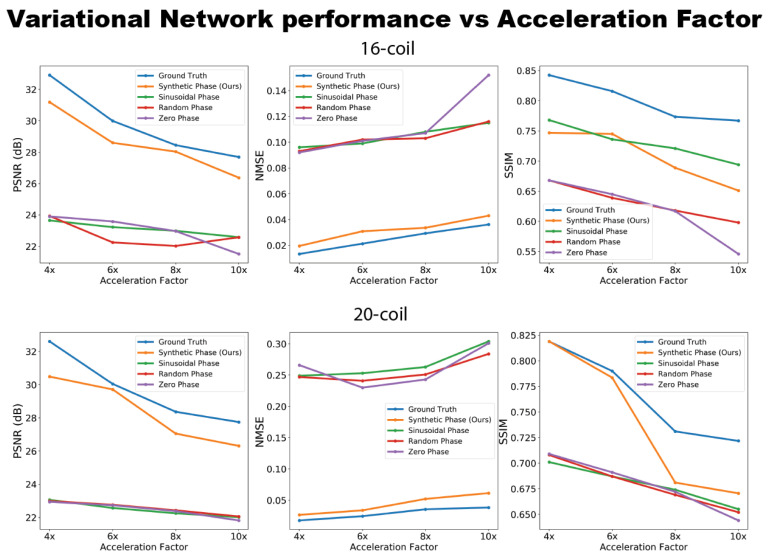
Performance of ground truth-trained and synthetically trained Variational Network reconstruction models at different acceleration factors for 16-coil and 20-coil datasets. At up to 10× acceleration factors, synthetically trained models show comparable performance to ground-truth trained models. These data are also shown in Appendix A.

## Data Availability

Not Applicable.

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
