# Peer review of "Synthesizing Complex-Valued Multicoil MRI Data from Magnitude-Only Images"

_bioengineering, 2023, doi:10.3390/bioengineering10030358_

Round 1
Reviewer 1 Report
Review: Synthesizing Complex Multicoil MRI Data from Magnitude-only Images
This paper proposes a method to syntetize MRI mage based on A GAN approach to obtain the image phase component and multichannel components from magnitude images, inspired in a image to image translation
Concerning paper organisation, it follows a general organisation, introducing slightly the problematic and and contextualisation, followed by the gan formulation for the phase generations and multi coil data generation, followed by the conducted experiments with different configurations and discussion.
The proposed architecture, explores the widely know PatchGAN with a 70X70 window patch to discriminate the patch images generations from the UNET and a Variational Network for final reconstruction.
As order of comparison, the addition of PSRN, NMSE and SSIM in a table in comparisons with other SOTA methods would be of great value.
While There is a deep study on the effects of number of coils, it would be helpful to see how the proposed pipeline compares to other methods in a summarised way, allowing to establish performance comparisons vs tradeoffs of the mentioned methods.
The plots only have a trade-off of the several PSNR, NMSE, SSIM on the acceleration factor, I suggest keep the acceleration factor at a value 4x or 8X and evaluate the overall performance as a extra study against other works.
Also, how the proposed pipeline performance on the faster challenge? The data was collected from there, so a submission to see where it ranks would be nice to see.
While the proposed research is coherent, however my concerns manly include:
1. The abstract and intro should empathise the problematic that the proposed work tries to address. More details can be introduced to empathise the scope of the work to the reader.
Other Issues:
- In some tables references, references are not clearly referenced in the text
- In some parts of the text, the sentences are not clear or missing some word connector, They are minor but can be fixed easily
- Conclusions more elaborated and discussed with a brother scenario of comparative analysis with pros and cons and more substance and compassion with other Sota would enrich the research.
Reviewer 2 Report
Review
Title: Synthesizing Complex Multicoil MRI Data from Magnitude-only Images
Authors: Nikhil Deveshwar, Abhejit Rajagopal, Sule Sahin, Efrat Shimron and Peder E. Z. Larson
In this manuscript, the authors present a method for synthesizing realistic MR images that are complex-valued and multi-channel from magnitude-only images. They used a conditional GAN based framework method in order to generate synthetic phase images from input magnitude images and ESPIRiT sensitivity maps derived from ground truth test data to generate multi-channel data.
In my opinion, this manuscript could be considered for publication, after the following minor changes:
1. The authors should put the paper in the journal format.
2. Furthermore, all the figures (images) clarity, resolution and text font size must be increased. The presented images are unclear and the font size it is too small which makes them difficult to read.
3. As a suggestion figure 1 should be arranged on vertically.
4. Please increase the dimension of each image. The predented images are too small and unclear.
5. Please see and refer:
a. https://doi.org/10.3390/pharmaceutics15010177
b. https://doi.org/10.1155/2013/587021
Reviewer 3 Report
This paper describes a generative model for synthesizing multicoil MRI data, from magnitude-only MRI images. The reason is that k-space/phase data is often important for quality reconstruction and various clinical applications, but such data is often discarded in practice, with only the magnitude data retained. As such, a conditional GAN framework was designed to generate synthetic phase images, from input magnitude images. These synthetic phase images are then compared against the true ground truth phase images, for physics-based reconstruction purposes via variational networks.
While the motivation and general workflow is sound, there may be some fundamental concerns for consideration:
1. Formulas relating the various MRI components might be provided & discussed (e.g. k-space image decomposable into real & imaginary components by Fourier; magnitude image = sqrt[(real^2)*(imag^2)]; phase image = arctan[imag/real])
2. Related to the above, while it is recognized that the phase image is not deterministic (i.e. fully defined by the magnitude image), it would appear to still be constrained by the known magnitude image via the above equations, particularly in the multi-coil data generation part of the workflow. It might be discussed if consistency was attempted to be enforced between the known magnitude and its synthetic phase image, in this framework.
3. In the Methods section, the discriminator is stated to be a PatchGAN. It might be explained as to why a PatchGAN was required, since the discriminator is merely required to try and classify whether an image is real or synthetic. Moreover, it might also be explained as to why discrimination was constrained to a 70x70 patch, rather than the full phase image.
4. In the Synthetic Phase Generation subsection, the two components of the conditional adversarial loss function might be explained in greater detail.
5. In the Evaluation section, it is stated that "each model was trained on synthetic and ground truth multicoil k-space with a 80/10/10... split". It might be clarified as to whether the 10% test split was on the true ground truth data for both the synthetic and ground truth models.
6. It might be considered to train a conventional CNN model (e.g. ResNet) to classify between synthetic (as generated by the trained cGAN model) and ground truth images, to check whether they can be empirically distinguished, and to what extent.
Round 2
Reviewer 1 Report
In a general way, the authors have answered my comments.
Reviewer 3 Report
We thank the authors for addressing our previous comments.